# Synthesis and Antibacterial Evaluation of *N*-phenylacetamide Derivatives Containing 4-Arylthiazole Moieties

**DOI:** 10.3390/molecules25081772

**Published:** 2020-04-12

**Authors:** Hui Lu, Xia Zhou, Lei Wang, Linhong Jin

**Affiliations:** State Key Laboratory Breeding Base of Green Pesticide and Agricultural Bioengineering, Key Laboratory of Green Pesticide and Agricultural Bioengineering, Ministry of Education, Guizhou University, Huaxi District, Guiyang 550025, China; luhui2624904231@163.com (H.L.); wanglei880328@163.com (L.W.)

**Keywords:** synthesis, thiazole, nematicidal activity, antibacterial activity, structure-activity relationship

## Abstract

A series of new *N*-phenylacetamide derivatives containing 4-arylthiazole moieties was designed and synthesized by introducing the thiazole moiety into the amide scaffold. The structures of the target compounds were confirmed by ^1^H-NMR, ^13^C-NMR and HRMS. Their in vitro antibacterial activities were evaluated against three kinds of bacteria—*Xanthomonas oryzae* pv. *Oryzae* (*Xoo*), *Xanthomonas axonopodis* pv. *Citri* (*Xac*) and *X.oryzae* pv*. oryzicola* (*Xoc*)—showing promising results. The minimum 50% effective concentration (EC_50_) value of *N-*(4-((4-(4-fluoro-phenyl)thiazol-2-yl)amino)phenyl)acetamide (**A**_1_) is 156.7 µM, which is superior to bismerthiazol (230.5 µM) and thiodiazole copper (545.2 µM). A scanning electron microscopy (SEM) investigation has confirmed that compound **A_1_** could cause cell membrane rupture of *Xoo*. In addition, the nematicidal activity of the compounds against *Meloidogyne incognita* (*M. incognita*) was also tested, and compound **A**_23_ displayed excellent nematicidal activity, with mortality of 100% and 53.2% at 500 μg/mL and 100 μg/mL after 24 h of treatment, respectively. The preliminary structure-activity relationship (SAR) studies of these compounds are also briefly described. These results demonstrated that phenylacetamide derivatives may be considered as potential leads in the design of antibacterial agents.

## 1. Introduction

*Xanthomonas oryzae* pv. *oryzae* and *Xanthomonas oryzae* pv. *oryzicola* parasitizing rice can cause serious degradation of rice quality and yield [1]. *Xanthomonas axonopodis* pv. *citri* mainly harms citrus fruits while causing a plant bacterial disease, citrus canker [2]. In recent decades, the long-term use of traditional bactericides has resulted in their large accumulation in the soil and resistance of pathogenic bacteria, reducing control capability. For example, bismerthiazol, one of the most widely used bactericides, has indicated low efficiency of 25.5% at a high dosage of 200 μg/mL [3]. Therefore, it is necessary to develop highly efficient and environmentally friendly bactericides to protect crops. Thiazole-containing aromatic heterocycles are reported to bind to certain proteins and receptors of bacteria and exhibit a wide range of biological activities such as insecticidal [4,5], bactericidal [6,7,8], herbicidal [9], antiviral [10] and fungicidal ones [11,12]. In addition, compounds with an amide structure have revealed various activities as antimicrobial [13,14,15] and nematicidal [16,17] agents. Based on the reported good performance of derivatives containing thiazole or amide groups, splicing them to get new antibacterial structures was presumed to be a reasonable and promising approach.

Attracted by these facts, we have carried out the preparation of a series of novel *N*-phenylacetamide derivatives **A**_1_–**A**_36_ and the subsequent evaluation of their biological activities against *Xoo*, *Xac* and *Xoc*. Their preliminary structure-activity relationships are discussed. Additionally, the nematicidal activity of all target compounds against *M. incognita* were also checked.

## 2. Results and Discussion

### 2.1. Chemistry

As shown, compounds *N*-substituted-2-amino-4-arylthiazoles **A**_1_–**A**_36_ were synthesized following Scheme 1. Starting from *p*-phenylenediamine (PPD), the synthesis involves firstly aniline protection, amide formation and deprotection to afford the 4-amino-*N*-phenylacetamide intermediates **2** which were then turned into isothiocyanates **3** and converted to thioureas **4**. Finally, **4** were condensed with different α-halocarbonyl compounds to give the target derivatives. The steps for each reaction described in Scheme 1 were elaborated as follows.

Specifically, the protection of PPD in first step utilized a *N*-tert-butyloxycarbonylation strategy [18,19,20]. By reacting with 0.5 equiv. of di-*tert*-butyl dicarbonate (BOC_2_O) in dichloromethane (DCM), PPD was converted into the corresponding *N*-BOC-amine [20]. The mono-*N*-BOC protected derivative (NH_2_-Ph-NH-BOC) **1** was purified by column chromatography to remove the di-*N*-BOC (BOC-NH-Ph-NH-BOC) side product. Then **1** was reacted with an acyl chloride (RCOCl) to produce the corresponding amide RCONH-Ph-NH-BOC, which was then acidified to produce the 4-amino-*N*-phenylacetamides **2**. Thereafter, **2** were converted to the corresponding isothiocyanates **3** via dithiocarbamates (generated by adding CS_2_ to Et_3_N) promoted by BOC_2_O and a catalytic amount of 4-dimethylaminopyridine (DMAP) [21]. BOC_2_O here, as reported, may desulfurize the corresponding dithiocarbamates during the isothiocyanate formation [21,22]. Then, various aryl thioureas **4** were prepared from isothiocyanates **3** upon reaction with ammonia [23,24]. Finally, thioureas **4** were condensed with diverse α-bromophenylethanones [24,25], synthesized separately via reaction of acetophenones with Br_2_ in DCM [24], to afford the 2-amino-1,3-thiazole heterocyclic skeleton, and the targeted compound *N*-phenylacetamide derivatives containing thiazole moieties were thus obtained. The structures of all target compounds were confirmed by ^1^H-NMR, ^13^C-NMR and HRMS. Among them, it was unexpectedly found that compounds **A**_4_, **A**_7_, **A**_11_, **A**_12_ have been previously reported [25].

### 2.2. In Vitro Antibacterial Activity 

The in vitro antibacterial activities of the target compounds against three phytopathogenic bacterial (*Xoo*, *Xac* and *Xoc*) were initially evaluated at the concentrations of 200 μg/mL and 100 μg/mL, respectively. As a comparison, the commercial bactericides bismerthiazol and thiodiazole copper served as positive controls (see preliminary screening results in the Appendix A).

Compounds that performed well in the initial tests were further tested to determine their EC_50_ values, with the results shown in Table 1. The EC_50_ values of compounds **A**_1_, **A**_4_, **A**_6_ against *Xoo* are 156.7, 179.2, 144.7 µM, which were significantly better than that of thiodiazole copper (545.2 µM). Moreover, compound **A**_4_ had the best inhibitory effect on *Xoc* with an EC_50_ value of 194.9 µM, which was slightly better than that of bismerthiazol (254.96 µM) and thiodiazole copper (607.5 µM). Compound **A**_4_ (281.2 µM) showed the best antibacterial activity against *Xac*, which was better than that of the commercial bactericide thiodiazole copper (476.52 µM).

### 2.3. Structure-Activity Relationship Analyses

With the results indicated in Table 1, a preliminary structure-activity relationship of the target compounds can be discussed. It can be observed that the type and position of the substituent R on the benzene ring had an important effect on the bactericidal activity of the target compounds. First of all, the type and position of F, Cl, Br, and CF_3_ at the 4-position of the benzene ring can increase the bactericidal activity of the compound, and the 3-position is not conducive to improving the bactericidal activity. For example, the order of the inhibitory effect of the target compound on *Xoo* bacteria follows the order **A**_1_ > **A**_2_, **A**_4_ > **A**_5_, confirming the above conclusion. In addition, the activity trend **A**_1_ (4-F) > **A**_4_ (4-Cl) > **A**_7_ (4-Br) reveals that a 4-F substituted benzene ring is the most helpful for conferring antibacterial activity. Secondly, comparing different types of groups at the same position on the benzene ring, the corresponding compounds with 4-R electron-withdrawing substituents have a higher bactericidal activity against *Xoo*, *Xac* and *Xoc* than 4-R electron-donating substituents such as **A**_1_ (4-F) > **A**_4_ (4-Cl) > **A**_11_ (4-CH_3_). The type of amide-linked sidechain could also have a significant impact on the bactericidal activity of the corresponding target compounds.

### 2.4. Scanning Electron Microscopy Studies

Due to its outstanding activity shown in Table 1, compounds **A**_1_ was further examined by SEM analysis to study the effect on *Xoo*. It can be observed that the cell membrane was damaged by the compound and the normal physiological functions of the cell would thus be affected. More importantly, this adverse effect becomes more severe with increasing compound concentration. For example, the surface of cells without compound treatment is smooth and the cell membrane is intact (Figure 1A). At the concentration of 100 μg/mL, a small part of the cell morphology appears abnormal (Figure 1B). When the concentration was increased to 200 μg/mL, most of the cell surfaces were deformed, with few surviving cells (Figure 1C). In summary, the inhibitory effect of compound **A**_1_ on *Xoo* was further clarified by the SEM images.

### 2.5. Nematicidal Biological Activities

The nematicidal activity results of the target compounds are summarized in Table 2. Unfortunately, most of the compounds showed poor activities against *M. incognita*, although compound **A**_23_, with its mortality rates of 100% at 500 μg/mL and 51.3% at 100 μg/mL which are comparable to the commercial nematicide avermectin (100% at 500 μg/mL and 71.8% at 100 μg/mL) in 24 h after treatment stuck out.

## 3. Experimental

### 3.1. Chemicals and Instruments

All reagents used for reactions were purchased from a commercial source (Aladdin Chemistry Co., Shanghai, China) and were of analytical grade with no further purification. Basic alumina oxide (200 to 300 mesh, Aladdin Chemistry Co.) was applied for purification of target compounds by column chromatography. Melting points of the target compounds were recorded on an XT-4B binocular microscope (Beijing Tech Instrument Co., Beijing, China). NMR spectra were obtained on a 400 MHz spectrometer (Bruker BioSpin AG, Fällanden, Switzerland) using TMS as internal standard and DMSO-*d_6_* as solvent. HRMS data were recorded on a Thermo Scientific Q Exactive system (Thermo Fisher Scientific, Waltham, MA, USA). Single crystal structure data were collected using a single crystal diffractometer (Gemini E, Oxford Instruments, Oxford, UK). SEM analysis was carried out with a FEI Nova NanoSEM 450 (FEI Company, Hillsboro, OR, USA).

### 3.2. General Synthetic Procedure for the Target Compounds

#### 3.2.1. Synthesis of Intermediate **1**

*p*-Phenylenediamine (PPD, 9.25 mmol, 1.0 equiv.) was dissolved in dry dichloromethane (50mL), degassed under N_2_ (g) and cooled in an ice bath. BOC_2_O (4.62 mmol, 0.50 equiv.) was added dropwise to the *p*-phenylenediamine solution using a disposable syringe. After the addition, the mixture was reacted at room temperature (R.T.) for 5 h, monitored by thin layer chromatography (TLC). After the reaction was complete, the mixture was purified by flash column chromatography (petroleum ether:ethyl acetate, 3:1) to obtain yellow solid compound intermediate **1** in a yield of 63%.

#### 3.2.2. Synthesis of Intermediates **2**

A mixture of **1** (1.0 equiv.) and an acid chloride (1.10 equiv.) in DCM (40mL) was stirred at 0 °C for 1 h. Then, excess 4 M HCl was added and stirred at R.T. for 10 min. The reaction progress was monitored by TLC. After completion of the reaction, a saturated Na_2_CO_3_ solution was added to adjust to 7.0 the pH of mixture which was then extracted with DCM (50 mL × 3). The organic phase was concentrated to give intermediate **2** as white solids in yields of 69–75%.

#### 3.2.3. Synthesis of Intermediates **3**

A mixture of intermediate **2** (12.64 mmol, 1.0 equiv.), triethylamine (Et_3_N, 13.87 mmol, 1.10 equiv.) and carbon disulfide (CS_2,_ 126.40 mmol, 10.0 equiv) in ethanol (50 mL) was stirred at R.T. for 2 h, then cooled down to 0 °C to add DMAP (0.38 mmol, 0.03 equiv.) and BOC_2_O (12.51 mmol, 0.99 equiv.). The mixture was stirred for about 2 h at room temperature till the reactants were completely consumed. The mixture was cooled to −10 °C overnight then filtered to give the isothiocyanates **3** in yields of 75–81%.

#### 3.2.4. Synthesis of the Intermediates **4**

The mixture of **3** (12.64 mmol, 1.0 equiv.) with excess ammonia solution in ethanol (30 mL) was stirred at R.T. The reaction was monitored by TLC using petroleum ether:ethyl acetate (1:1). The mixture was filtered through a Celite pad to afford the thioureas **4** in approximately 100% yield.

#### 3.2.5. Synthesis of the α-Bromophenylethanone Intermediates

Br_2_ (2.2 mmol, 1.10 equiv.) was added slowly to a DCM solution of substituted acetophenone (2.0 mmol, 1.0 equiv.) and the mixture was stirred at R.T. for 5 h, after which the reaction mixture was tpoured into distilled water and extracted with DCM three times. The organic phases were combined and the solvent removed under reduced pressure to obtain the α-bromophenylethanones in yields of 65–83%.

#### 3.2.6. Synthesis of the Target Compounds **A**

A mixture of α-bromophenylethanone (2.05 mmol, 1.0 equiv.) and intermediate **4** (2.25 mmol, 1.10 equiv.) was dissolved in ethanol (20 mL) and then heated at reflux for 5 h. Upon completion of the reaction the liquid was filtered and the solid formed was washed with saturated potassium carbonate solution. The crude product was purified by column chromatography on basic alumina to give the various target compounds in yields ranging from 55% to 94%, respectively.

*N-(4-((4-(4-Fluorophenyl)thiazol-2-yl)amino)phenyl)acetamide* (**A**_1_): yellow solid; yield 57%; mp: 201.0–202.0 °C; ^1^H-NMR δ 10.19 (s, 1H), 9.85 (s, 1H), 7.96 (m, 2H, Ar-H), 7.63 (d, *J* = 8.9 Hz, 2H, Ar-H), 7.54 (d, *J* = 8.9 Hz, 2H, Ar-H), 7.27 (d, *J* = 8.5 Hz, 2H, Ar-H), 7.24 (s, 1H, thiazole-H), 2.03 (s, 3H, CH_3_). ^13^C-NMR δ 168.29, 163.84, 163.28, 160.85, 149.48, 137.11, 133.80, 131.65 (d, *J* = 2.9 Hz), 128.14 (d, *J* = 8.1 Hz), 120.35, 117.76, 116.05, 115.83, 102.72, 24.35. HRMS (ESI): *m*/*z* [M + H]^+^ calculated for C_17_H_15_FN_3_OS: 328.09144; found: 328.09097.

*N-(4-((4-(3-Fluorophenyl)thiazol-2-yl)amino)phenyl)acetamide* (**A**_2_): yellow solid; yield 82%; mp: 162.4–163.2 °C; ^1^H-NMR δ 10.16 (s, 1H, NH), 9.81 (s, 1H,NH), 7.91 (d, *J* = 8.6 Hz, 2H, Ar-H), 7.59 (d, *J* = 9.0 Hz, 2H, Ar-H), 7.51 (d, *J* = 9.0 Hz, 2H, Ar-H), 7.45 (d, *J* = 8.6 Hz, 2H, Ar-H), 7.33 (s, 1H, thiazole-H), 1.99 (s, 3H, CH_3_). ^13^C-NMR δ 168.26, 163.65, 150.60, 137.22 (d, *J* = 4.8 Hz), 133.71, 132.41, 129.67, 126.10, 120.35, 117.70, 102.07, 24.35, 21.33. HRMS (ESI): *m*/*z* [M + H]^+^ calculated for C_17_H_15_FN_3_OS: 328.09144; found: 328.09094.

*N-(4-(4-(3,4-Difluorophenyl)thiazol-2-ylamino)phenyl)acetamide* (**A**_3_): yellow solid; yield 81%; mp: 228.8–229.3 °C; ^1^H-NMR δ 10.21 (s, 1H), 9.85 (s, 1H), 7.91–7.87 (m, 1H, Ar-H), 7.77–7.72 (m, 1H, Ar-H), 7.59 (d, *J* = 9.0 Hz, 2H, Ar-H), 7.52 (d, *J* = 9.0 Hz, 2H, Ar-H), 7.45 (d, *J* = 10.6 Hz, 1H, Ar-H), 7.36 (s, 1H, thiazole-H), 2.00 (s, 3H, CH_3_). ^13^C-NMR δ 168.2, 163.98, 148.82, 137.29, 136.99, 133.94, 131.35, 130.62, 128.56, 125.16, 122.56, 120.37, 117.91, 104.64, 24.35. HRMS (ESI): *m*/*z* [M + H]^+^ calculated for C_17_H_14_F_2_N_3_OS: 346.08202; found: 346.08136.

*N-(4-(4-(4-Chlorophenyl)thiazol-2-ylamino)phenyl)acetamide* (**A**_4_): yellow solid; yield 87%; mp: 210.5–211.2 °C; ^1^H-NMR δ 10.16 (s, 1H, NH), 9.81 (s, 1H, NH), 7.93–7.88 (m, 2H, Ar-H), 7.59 (d, *J* = 9.0 Hz, 2H, Ar-H), 7.51 (d, *J* = 8.6 Hz, 2H, Ar-H), 7.45 (d, *J* = 8.6 Hz, 2H, Ar-H), 7.33 (s, 1H, thiazole-H), 1.99 (s, 3H, CH_3_). ^13^C-NMR δ 168.34, 163.98, 149.37, 137.12, 133.94, 132.43, 129.19, 127.93, 120.41, 117.89, 103.90, 24.42. HRMS (ESI): *m*/*z* [M + H]^+^ calculated for C_17_H_15_ClN_3_OS: 344.06189; found: 344.06134.

*N-(4-(3-(4-Chlorophenyl)thiazol-2-ylamino)phenyl)acetamide* (**A**_5_): yellow solid; yield 88%; mp: 179.7–180.3 °C; ^1^H-NMR δ 10.22 (s, 1H, NH), 9.86 (s, 1H, NH), 7.78 (d, *J* = 7.9 Hz, 1H, Ar-H), 7.71 (d, *J* = 9.8 Hz, 1H, Ar-H), 7.65–7.59 (m, 2H, Ar-H), 7.55 (d, *J* = 9.0 Hz, 2H, Ar-H), 7.48 (d, *J* = 6.2 Hz, 1H, Ar-H), 7.44 (s, 1H, thiazole-H), 7.14 (d, *J* = 2.5 Hz, 1H, Ar-H), 2.03 (s, 3H, CH_3_). ^13^C-NMR δ 168.35, 163.90, 148.71, 137.31, 137.03, 133.93, 131.37, 130.58, 128.51, 125.15, 122.55, 120.28, 117.79, 104.70, 24.34. HRMS (ESI): *m*/*z* [M + H]^+^ calculated for C_17_H_15_ClN_3_OS: 344.06189; found: 344.06140.

*N-(4-((4-(3,4-Dichlorophenyl)thiazol-2-yl)amino)phenyl)acetamide* (**A**_6_): yellow solid; yield 93%; mp: 179.3–181.7 °C; ^1^H-NMR δ 10.55 (s, 1H, NH), 10.15 (s, 1H, NH), 8.14 (d, *J* = 2.0 Hz, 1H, Ar-H), 7.93 (m, 1H, Ar-H), 7.70 (d, *J* = 8.5 Hz, 1H, Ar-H), 7.67 (d, *J* = 9.1 Hz, 2H, Ar-H), 7.61 (d, *J* = 9.1 Hz, 2H, Ar-H), 7.54 (s, 1H, thiazole-H), 2.06 (s, 3H, CH_3_). ^13^C-NMR δ 168.37, 164.00, 147.82, 136.97, 135.60, 133.99, 131.86, 131.37, 130.05, 127.58, 126.30, 120.26, 117.80, 105.37, 24.34. HRMS (ESI): *m*/*z* [M + H]^+^ calculated for C_17_H_14_Cl_2_N_3_OS: 378.02291; found: 378.02249.

*N-(4-((4-(4-Bromophenyl)thiazol-2-yl)amino)phenyl)acetamide* (**A**_7_): white solid; yield 75%; mp: 209.6–210.4 °C; ^1^H-NMR δ 10.19 (s, 1H, NH), 9.83 (s, 1H, NH), 7.84 (m, 2H, Ar-H), 7.60 (d, *J* = 9.1 Hz, 2H, Ar-H), 7.58 (d, *J* = 4.4 Hz, 2H, Ar-H), 7.52 (d, *J* = 8.8 Hz, 2H, Ar-H), 7.33 (s, 1H, thiazole-H), 2.00 (s, 3H, CH_3_). ^13^C-NMR δ 168.36, 163.96, 149.40, 137.11, 134.27, 133.91, 132.10, 128.24, 121.04, 120.40, 117.87, 103.98, 24.43. HRMS (ESI): *m*/*z* [M + H]^+^ calculated for C_17_H_15_BrN_3_OS: 388.01137; found: 388.01102.

*N-(4-((4-(3-Bromophenyl)thiazol-2-yl)amino)phenyl)acetamide* (**A**_8_): yellow solid; yield 85%; mp: 202.8–203.9 °C; ^1^H-NMR δ 10.38 (s, 1H, NH), 10.01 (s, 1H, NH), 8.04 (t, *J* = 1.8 Hz, 1H, Ar-H), 7.90 (d, *J* = 7.9 Hz, 1H, Ar-H), 7.62-7.57 (m, 2H, Ar-H), 7.56-7.52 (m, 2H, Ar-H), 7.46 (d, *J* = 6.9 Hz, 1H, Ar-H), 7.42 (s, 1H, thiazole-H), 7.36 (t, *J* = 7.9 Hz, 1H, Ar-H), 2.00 (s, 3H,CH_3_). ^13^C-NMR δ 168.29, 163.98, 148.82, 137.29, 136.99, 133.94, 131.35, 130.62, 128.56, 125.16, 122.56, 120.37, 117.91, 104.64, 24.35. HRMS (ESI): *m*/*z* [M + H]^+^ calculated for C_17_H_15_BrN_3_OS: 388.01137; found: 388.01102.

*N-(4-(4-(4-(Trifluoromethyl)phenyl)thiazol-2-ylamino)phenyl)acetamide* (**A**_9_): yellow solid; yield 69%; mp: 253.2–254.2 °C; ^1^H-NMR δ 10.29 (s, 1H, NH), 9.89 (s, 1H, NH), 8.14 (d, *J* = 8.3 Hz, 2H, Ar-H), 7.73-7.79 (d, *J* = 7.8 Hz, 2H, Ar-H), 7.65 (m, 2H, Ar-H), 7.58 (d, *J* = 2.2 Hz, 2H, Ar-H), 7.56 (s, 1H, thiazole-H), 2.04 (s, 3H, CH_3_). ^13^C-NMR δ 168.30, 164.01, 148.99, 138.67, 136.97, 133.93, 128.14, 127.82, 126.87, 126.38 (d, *J* = 59.5 Hz), 123.50, 120.30, 117.85, 105.87, 24.36. HRMS (ESI): *m*/*z* [M + H]^+^ calculated for C_18_H_15_F_3_N_3_OS: 378.08824; found: 378.08765.

*N-(4-((4-(3-(Trifluoromethyl)phenyl)thiazol-2-yl)amino)phenyl)acetamide* (**A**_10_): yellow solid; yield 87%; mp: 179.3–180.7 °C; ^1^H-NMR δ 10.31 (s, 1H, NH), 9.93 (s, 1H,NH), 8.26 (d, *J* = 11.0 Hz, 2H, Ar-H), 7.69 (d, *J* = 6.9 Hz, 2H, Ar-H), 7.67–7.62 (m, 2H, Ar-H), 7.60 (d, *J* = 9.1 Hz, 2H, Ar-H), 7.58 (s, 1H, thiazole-H), 2.07 (s, 3H, CH_3_). ^13^C-NMR δ 168.39, 164.18, 148.81, 136.98, 135.88, 133.98, 130.06 (d, *J* = 33.3Hz), 129.52, 128.79, 126.09, 124.38 (d, *J* = 3.6 Hz), 123.38, 122.35 (d, *J* = 3.7 Hz), 120.38, 117.97, 104.95, 24.32. HRMS (ESI): *m*/*z* [M + H]^+^ calculated for C_17_H_15_F_3_N_3_OS: 378.08824; found: 378.08765.

*N-(4-(4-p-Tolylthiazol-2-ylamino)phenyl)acetamide* (**A**_11_): yellow solid; yield 89%; mp: 207.2–208.1 °C; ^1^H-NMR δ 10.15 (s, 1H, NH), 9.85 (s, 1H, NH), 7.81 (d, *J* = 8.0 Hz, 2H, Ar-H), 7.63 (d, *J* = 8.9 Hz, 2H, Ar-H), 7.54 (d, *J* = 8.9 Hz, 2H, Ar-H), 7.23 (d, *J* = 8.1 Hz, 2H, Ar-H), 7.21 (s, 1H), 2.33 (s, 3H, CH_3_), 2.03 (s, 3H, CH_3_). ^13^C-NMR δ 168.26, 163.66, 150.61, 137.23, 133.72, 132.41, 129.67, 126.10, 120.36, 117.71, 102.06, 24.35, 21.32; HRMS (ESI): *m*/*z* [M + H]^+^ calculated for C_18_H_18_N_3_OS: 324.11651; found: 324.11597.

*N-(4-((4-Phenylthiazol-2-yl)amino)phenyl)acetamide* (**A**_12_): yellow solid; yield 71%; mp: 174.7–175.2 °C; ^1^H-NMR δ 10.19 (s, 1H, NH), 9.88 (s, 1H, NH), 7.9–7.86 (m, 2H, Ar-H), 7.66 (d, *J* = 9.0 Hz, 2H, Ar-H), 7.57 (d, *J* = 9.0 Hz, 2H, Ar-H), 7.43 (t, *J* = 7.6 Hz, 2H, Ar-H), 7.31 (t, *J* = 7.3 Hz, 1H, Ar-H), 7.27 (s, 1H, thiazole-H), 2.04 (s, 3H, CH_3_). ^13^C-NMR δ 168.38, 163.79, 150.57, 137.22, 135.03, 133.74, 129.11, 128.01, 126.15, 120.44, 117.76, 102.97, 24.32. HRMS (ESI): *m*/*z* [M + H]^+^ calculated for C_17_H_16_N_3_OS: 310.10086; found: 310.10034.

*N-(4-((4-(4-Fluorophenyl)thiazol-2-yl)amino)phenyl)propionamide* (**A**_13_): white solid; yield 85%; mp: 186.5–187.4 °C; ^1^H-NMR δ 10.24 (s, 1H, NH), 9.80 (s, 1H, NH), 7.95 (s, 1H, thiazole-H), 7.93–7.87 (m, 1H, Ar-H), 7.64–7.60 (m, 2H, Ar-H),7.60–7.54 (m, 2H, Ar-H), 7.50–7.41 (m, 2H, Ar-H), 7.37 (dd, *J* = 5.9, 0.9 Hz, 1H), 2.31 (q, *J* = 7.6 Hz, 2H, CH_2_), 1.10 (t, *J* = 7.6 Hz, 3H, CH_3_). ^13^C-NMR δ 172.03, 164.03, 148.82, 137.30, 136.93, 134.00, 131.34, 130.62, 128.58, 125.14, 122.55, 120.41, 117.95, 104.62, 29.91, 10.25. HRMS (ESI): *m*/*z* [M + H]^+^ calculated for C_18_H_17_FN_3_OS: 342.10709; found: 342.10617.

*N-(4-((4-(3-Fluorophenyl)thiazol-2-yl)amino)phenyl)propionamide* (**A**_14_): white solid; yield 83%; mp: 230.2–231.1 °C; ^1^H-NMR δ 10.18 (s, 1H, NH), 9.76 (s, 1H, NH), 7.74 (d, *J* = 7.8 Hz, 1H, Ar-H), 7.67 (d, *J* = 10.6 Hz, 1H, Ar-H), 7.59 (d, *J* = 8.8 Hz, 2H, Ar-H), 7.54 (d, *J* = 8.9 Hz, 2H,Ar-H), 7.44 (dd, *J* = 14.3, 7.8 Hz, 1H, Ar-H), 7.39 (s, 1H, thiazole-H), 7.11 (dd, *J* = 11.7, 5.4 Hz, 1H, Ar-H), 2.27 (q, *J* = 7.5 Hz, 2H, CH_2_), 1.06 (t, *J* = 7.5 Hz, 3H, CH_3_). ^13^C-NMR δ 172.04, 164.21, 163.89, 161.80, 149.24, 137.40, 136.95, 133.91, 131.13, 122.23, 120.41, 117.86, 114.77, 112.48, 104.53, 29.90, 10.27. HRMS (ESI): *m*/*z* [M + H]^+^ calculated for C_18_H_17_FN_3_OS: 342.10709; found: 342.10617.

*N-(4-((4-(3,4-Difluorophenyl)thiazol-2-yl)amino)phenyl)propionamide* (**A**_15_): white solid; yield 69%; mp: 230.4–230.8 °C; ^1^H-NMR δ 10.18 (s, 1H, NH), 9.75 (s, 1H, NH), 7.92-7.87 (m, 1H, Ar-H), 7.75 (dd, 1H, *J* = 8.6, 4.3 Hz, Ar-H), 7.61-7.56 (m, 2H, Ar-H), 7.53 (dd, *J* = 9.2, 2.3 Hz, 2H, Ar- H), 7.46 (d, *J* = 10.6, 1H, Ar-H), 7.37 (s, 1H, thiazole-H), 2.27 (q, *J* = 7.6 Hz, 2H, CH_2_), 1.05 (t, *J* = 7.6 Hz, 3H, CH_3_). ^1^^3^C-NMR δ 172.00, 164.00, 148.37, 136.89, 133.96, 122.92, 120.39, 118.24 (d, *J* = 17.2 Hz), 117.89, 114.87 (d, *J* = 18.4 Hz), 104.16, 29.90, 10.27. HRMS (ESI): *m*/*z* [M + H]^+^ calculated for C_18_H_16_F_2_N_3_OS: 360.09767; found: 360.09653.

*N-(4-((4-(4-Chlorophenyl)thiazol-2-yl)amino)phenyl)propionamide* (**A**_16_): white solid; yield 90%; mp: 269.3–270.1 °C; ^1^H-NMR δ 10.24 (s, 1H, NH), 9.80 (s, 1H, NH), 7.97–7.84 (m, 2H, Ar-H), 7.62 (d, *J* = 9.2 Hz, 2H, Ar-H), 7.58 (d, *J* = 9.2 Hz, 2H, Ar-H), 7.50–7.42 (m, 2H, Ar-H), 7.33 (s, 1H, thiazole-H), 2.21 (q, *J* = 7.6 Hz, 2H, CH_2_), 1.05 (t, *J* = 7.6 Hz, 3H, CH_3_). ^13^C-NMR δ 172.07, 163.93, 149.32, 137.03, 133.94, 132.41, 129.20, 127.92, 120.38, 117.84, 103.90, 29.96, 10.36. HRMS (ESI): *m*/*z* [M + H]^+^ calculated for C_18_H_17_ClN_3_OS: 358.07754; found: 358.07639.

*N-(4-((4-(3-Chlorophenyl)thiazol-2-yl)amino)phenyl)propionamide* (**A**_17_): white solid; yield 82%; mp: 223.5–224.6 °C; ^1^H-NMR) δ 10.24 (s, 1H, NH), 9.80 (s, 1H, NH), 7.95 (s, 1H, thiazole-H), 7.90 (d, *J* = 7.8 Hz, 1H, Ar-H), 7.62 (d, *J* = 9.2 Hz, 2H, Ar-H), 7.58 (d, *J* = 9.2 Hz, 2H, Ar-H), 7.52–7.42 (m, 2H, Ar-H), 7.37 (dd, *J* = 8.0, 1.1 Hz, 1H, Ar-H), 2.31 (q, *J* = 7.6 Hz, 2H, CH2), 1.10 (t, *J* = 7.6 Hz, 3H, CH3). ^13^C-NMR δ 172.08, 164.05, 149.00, 137.05, 134.02, 131.13, 127.80, 125.74, 124.84, 120.46, 117.97, 104.70, 29.97, 10.34. HRMS (ESI): *m*/*z* [M + H]^+^ calculated for C_18_H_17_ClN3OS: 358.07754; found: 358.07639.

*N-(4-((4-(3,4-Dichlorophenyl)thiazol-2-yl)amino)phenyl)propionamide* (**A**_18_): white solid; yield 77%; mp: 258.1–259.0 °C; ^1^H-NMR δ 10.27 (s, 1H, NH), 9.80 (s, 1H, NH), 8.10 (d, *J* = 2.0 Hz, 1H, Ar-H), 7.88 (dd, *J* = 8.4, 2.1 Hz, 1H, Ar-H), 7.66 (d, *J* = 8.5 Hz, 1H, Ar-H), 7.57 (d, *J* = 9.2 Hz, 2H, Ar-H), 7.54 (d, *J* = 9.2 Hz, 2H, Ar-H), 7.50 (s, 1H, thiazole-H), 2.27 (q, *J* = 7.5 Hz, 2H, CH_2_), 1.05 (t, *J* = 7.6 Hz, 3H, CH_3_). ^13^C-NMR δ 172.12, 164.15, 147.98, 136.92, 135.65, 134.11, 131.95, 131.44, 130.17, 127.68, 126.36, 120.41, 118.01, 105.37, 29.96, 10.34. HRMS (ESI): *m*/*z* [M + H]^+^ calculated for C_18_H_16_Cl_2_N_3_OS: 392.03856; found: 392.03856 392.03784.

*N-(4-(4-(4-Bromophenyl)thiazol-2-ylamino)phenyl)propionamide* (**A**_19_): white solid; yield 65%; mp: 219.5–220.8 °C; ^1^H-NMR δ 10.17 (s, 1H, NH), 9.75 (s, 1H, NH), 7.84 (d, *J* = 8.5 Hz, 2H,), 7.61–7.58 (m, 2H), 7.58 (d, *J* = 2.0 Hz, 2H, Ar-H), 7.53 (d, *J* = 9.0 Hz, 2H, Ar-H), 7.34 (s, 1H, thiazole-H), 2.27 (q, *J* = 7.6 Hz, 2H, CH_2_), 1.05 (t, *J* = 7.6 Hz, 3H, CH_3_). ^13^C-NMR δ 172.07, 163.99, 149.39, 137.03, 134.28, 133.97, 132.10, 128.24, 121.03, 120.41, 117.89, 103.98, 29.97, 10.34. HRMS (ESI): *m*/*z* [M + H]^+^ calculated for C_18_H_17_BrN_3_OS: 402.02702; found: 402.02594.

*N-(4-(4-(3-Bromophenyl)thiazol-2-ylamino)phenyl)propionamide* (**A**_20_): white solid; yield 94%; mp: 238.4–239.1 °C; ^1^H-NMR δ 10.21 (s, 1H, NH), 9.79 (s, 1H, NH), 8.08 (t, *J* = 1.7 Hz, 1H, Ar-H), 7.93 (d, *J* = 7.8 Hz, 1H, Ar-H), 7.61 (d, *J* = 9.2 Hz, 2H, Ar-H), 7.57 (d, *J* = 9.2 Hz, 2H, Ar-H), 7.50 (dd, *J* = 8.0, 0.9 Hz, 1H, Ar-H), 7.45 (s, 1H, thiazole-H), 7.40 (t, J = 7.9 Hz, 1H, Ar-H), 2.31 (q, J = 7.5 Hz, 2H, CH_2_), 1.09 (t, *J* = 7.6 Hz, 3H, CH_3_). ^13^C-NMR δ 172.06, 164.02, 148.81, 137.29, 136.93, 133.98, 131.35, 130.62, 128.57, 125.13, 122.56, 120.41, 117.94, 104.62, 40.58, 40.37, 40.16, 39.95, 39.75, 39.54, 39.33, 29.91, 10.26. HRMS (ESI): *m*/*z* [M + H]^+^ calculated for C_18_H_17_BrN_3_OS: 402.02702; found: 402.02606.

*N-(4-((4-(4-(Trifluoromethyl)phenyl)thiazol-2-yl)amino)phenyl)propionamide* (**A**_21_): white solid; yield 82%; mp: 242.1–243.0 °C; ^1^H-NMR δ 10.23 (s, 1H, NH), 9.76 (s, 1H, NH), 8.22–8.16 (m, 2H, Ar-H), 7.64 (d, *J* = 5.9 Hz, 2H, Ar-H), 7.60-7.55 (m, 2H, Ar-H), 7.54 (d, *J* = 2.9 Hz, 2H, Ar-H), 7.53 (s, 1H, thiazole-H), 2.27 (q, *J* = 7.5 Hz, 2H, CH_2_), 1.05 (t, *J* = 7.6 Hz, 3H, CH_3_). ^13^C-NMR δ 172.02, 164.18, 148.80, 136.87, 135.90, 134.07, 130.07 (d, 17.6 Hz), 126.10, 124.37, 123.39, 122.35, 120.3, 117.98, 104.97, 29.90, 10.25. HRMS (ESI): *m*/*z* [M + H]^+^ calculated for C_19_H_17_F_3_N_3_OS: 392.10389; found: 392.10281.

*N-(4-((4-(3-(Trifluoromethyl)phenyl)thiazol-2-yl)amino)phenyl)isobutyramide* (**A**_22_): yellow solid; yield 86%; mp: 168.2–168.9 °C; ^1^H-NMR δ 10.24 (s, 1H, NH), 9.77 (s, 1H, NH), 8.21-8.17 (m, 2H, Ar-H), 7.64 (d, *J* = 7.1 Hz, 2H), 7.58 (d, *J* = 9.1 Hz, 2H, Ar-H), 7.55 (d, *J* = 2.3 Hz, 2H, Ar-H), 7.53 (s, 1H, thiazole-H), 2.27 (q, *J* = 7.5 Hz, 2H, CH_2_), 1.05 (t, *J* = 7.6 Hz, 3H, CH_3_). ^13^C-NMR δ 172.02, 164.18, 148.80, 136.87, 135.90, 134.07, 130.33, 129.97, 126.10, 124.37, 123.39, 122.35, 120.35, 117.98, 104.97, 29.90, 10.25. HRMS (ESI): *m*/*z* [M + H]^+^ calculated for C_19_H_16_F_3_N_3_OS: 392.10389; found: 392.10266.

*N-(4-((4-(p-Tolyl)thiazol-2-yl)amino)phenyl)propionamide* (**A**_23_): white solid; yield 79%; mp: 286.4–287.5 °C; ^1^H-NMR (400 MHz, DMSO-*d_6_*) δ 10.23 (s, 1H, NH), 9.85 (s, 1H, NH), 7.81 (d, *J* = 8.0 Hz, 2H, Ar-H), 7.66 (d, *J* = 9.0 Hz, 2H, Ar-H), 7.59 (d, *J* = 9.0 Hz, 2H, Ar-H), 7.24 (d, *J* = 8.0 Hz, 2H, Ar-H), 7.21 (s, 1H, Thiazole-H), 2.34 (d, *J* = 8.1 Hz, 3H, CH_3_), 2.30 (d, *J* = 7.5 Hz, 2H, CH_2_), 1.10 (t, *J* = 7.6 Hz, 3H, CH_3_). ^13^C-NMR (100 MHz, DMSO-*d_6_*) δ 172.01, 163.67, 150.59, 137.19, 133.78, 132.42, 129.67, 126.09, 120.36, 117.70, 102.04, 29.91, 21.32, 10.30. HRMS (ESI): *m*/*z* [M + H]^+^ calculated for C_19_H_20_N_3_OS: 338.13216; found: 338.13129.

*N-(4-((4-Phenylthiazol-2-yl)amino)phenyl)propionamide* (**A**_24_): white solid; yield 87%; mp: 187.5–187.9 °C; ^1^H-NMR δ 10.16 (s, 1H, NH), 9.76 (s, 1H, NH), 7.89 (dd, *J* = 8.2, 1.1 Hz, 2H, Ar-H), 7.61 (d, *J* = 9.0 Hz, 2H, Ar-H), 7.54 (d, *J* = 9.0 Hz, 2H, Ar-H), 7.39 (d, *J* = 7.7 Hz, 2H, Ar-H), 7.28 (dd, *J* = 10.5, 4.3 Hz, 1H, Ar-H), 7.26 (s, 1H, thiazole-H), 2.28 (q, *J* = 7.5 Hz, 2H, CH_2_), 1.06 (t, *J* = 7.6 Hz, 3H, CH_3_). ^13^C-NMR δ 171.99, 163.78, 150.55, 137.11, 135.04, 133.82, 129.11, 128.00, 126.15, 120.39, 117.76, 102.98, 29.91, 10.28. HRMS (ESI): *m*/*z* [M + H]^+^ calculated for C_18_H_20_N_3_OS: 324.11651; found: 324.11514.

*N-(4-(4-(4-Fluorophenyl)thiazol-2-ylamino)phenyl)isobutyramide* (**A**_25_): yellow solid; yield 74%; mp: 172.4–173.9 °C; ^1^H-NMR δ 10.16 (s, 1H, NH), 9.72 (s, 1H, NH), 7.92 (m, 2H, Ar-H), 7.62–7.57 (m, 2H, Ar-H), 7.56–7.51 (m, 2H, Ar-H), 7.23 (d, *J* = 4.4 Hz, 2H, Ar-H), 7.20 (s, 1H, thiazole-H), 2.59–2.50 (m, 1H), 1.07 (d, *J* = 6.8 Hz, 6H, CH_3_). ^13^C-NMR δ 175.27, 163.94, 163.29, 160.86, 149.50, 137.08, 133.91, 131.67 (d, *J* = 3.1 Hz), 128.13 (d, *J* = 8.0 Hz), 120.53, 117.81, 116.03, 115.81, 102.70, 56.51, 35.30, 20.04. HRMS (ESI): *m*/*z* [M + H]^+^ calculated for C_19_H_19_FN_3_OS: 356.12274; found: 356.12180.

*N-(4-((4-(3-Fluorophenyl)thiazol-2-yl)amino)phenyl)isobutyramide* (**A**_26_): yellow solid; yield 58%; mp: 230.9–232.0 °C; ^1^H-NMR δ 10.18 (s, 1H, NH), 9.72 (s, 1H, NH), 7.74 (d, *J* = 7.9 Hz, 1H, Ar-H), 7.67 (d, *J* = 9.8 Hz, 1H, Ar -H), 7.62–7.56 (m, 2H, Ar -H), 7.57–7.52 (m, 2H, Ar -H),, 7.44 (m, 1H, Ar-H), 7.40 (s, 1H, thiazole-H), 7.10 (m, 1H, Ar-H), 2.54 (m, 1H, CH), 1.07 (d, *J* = 6.8 Hz, 6H, CH_3_). ^13^C-NMR δ 175.24, 164.21, 163.91, 161.81, 149.25 (d, *J* = 2.8 Hz), 137.41 (d, *J* = 8.2 Hz), 136.98, 133.98, 131.13 (d, *J* = 8.5 Hz), 122.23, 120.54, 117.87, 114.66 (d, *J* = 21.3 Hz), 112.72, 112.49, 104.52, 35.29, 20.05. HRMS (ESI): *m*/*z* [M + H]^+^ calculated for C_19_H_19_FN_3_OS: 356. 12274; found: 356 12149.

*N-(4-((4-(3,4-Difluorophenyl)thiazol-2-yl)amino)phenyl)isobutyramide* (**A**_27_): yellow solid; yield 69%; mp: 190.6–191.6 °C; ^1^H-NMR δ 10.22 (s, 1H, NH), 9.76 (s, 1H, NH), 7.93 (dd, *J* = 11.2, 6.9 Hz, 1H, Ar-H), 7.78 (dd, *J* = 8.7, 4.3 Hz, 1H, Ar-H), 7.61 (d, *J* = 9.3 Hz, 2H, Ar-H), 7.57 (d, *J* = 9.3 Hz, 2H, Ar-H), 7.50 (d, *J* = 8.6 Hz, 1H, Ar-H), 7.41 (s, 1H, thiazole-H), 2.59 (m, 1H, CH), 1.10 (d, *J* = 6.8 Hz, 6H, CH_3_). ^13^C-NMR δ 175.31, 164.06, 151.24, 150.45, 148.87 (d, *J* = 12.7 Hz), 148.36, 148.06 (d, *J* = 12.7 Hz), 136.95, 133.99, 132.71, 122.89, 120.57, 118.22 (d, *J* = 17.1 Hz), 117.92, 114.86 (d, *J* = 18.3 Hz), 104.13, 35.29, 20.03. HRMS (ESI): *m*/*z* [M + H]^+^ calculated for C_19_H_18_N_3_OF_2_S: 374.11332; found: 374.11191.

*N-(4-((4-(4-Chlorophenyl)thiazol-2-yl)amino)phenyl)isobutyramide* (**A**_28_): yellow solid; yield 91%; mp: 264.5–265.7 °C; ^1^H-NMR δ 10.15 (s, 1H, NH), 9.72 (s,1H, NH), 7.88 (m, 2H, Ar-H) 7.65-7.57 (m, 2H, Ar-H), 7.57–7.52 (m, 2H, Ar-H), 7.40 (m, 2H, Ar-H), 7.28 (d, *J* = 7.3 Hz, 1H, Ar-H), 7.26 (s, 1H, thiazole-H), 2.60–2.49 (m, 1H, CH), 1.11 (d, *J* = 6.8 Hz, 6H, CH_3_). ^13^C-NMR δ 175.28, 163.92, 149.22, 137.04, 133.92, 132.33, 129.12, 127.84, 120.43, 117.77, 103.83, 35.30, 20.07. HRMS (ESI): *m*/*z* [M + H]^+^ calculated for C_19_H_19_ClN_3_OS: 372.09319; found: 372.09207.

*N-(4-((4-(3-Chlorophenyl)thiazol-2-yl)amino)phenyl)isobutyramide* (**A**_29_): yellow solid; yield 91%; mp: 234.4–235.7 °C; ^1^H-NMR δ 10.20 (s, 1H, NH), 9.73 (s, 1H, NH), 7.91 (t, *J* = 1.8 Hz, 1H, Ar-H), 7.86 (dd, *J* = 7.8, 0.9 Hz, 1H, Ar-H), 7.61–7.56 (m, 2H, Ar-H), 7.56–7.52 (m, 2H, Ar-H), 7.46–7.39 (m, 2H, Ar-H), 7.33 (dd, *J* = 8.0, 1.1 Hz, 1H, thiazole-H), 2.55 (m, 1H,CH), 1.07 (d, *J* = 6.8 Hz, 6H,CH_3_). ^13^C-NMR δ 175.25, 163.99, 148.93, 136.99, 133.98, 131.05, 127.73, 125.69, 124.76, 120.51, 117.90, 104.64, 35.30, 20.06. HRMS (ESI): *m*/*z* [M + H]^+^ calculated for C_19_H_19_ClN_3_OS: 372.09319; found: 372.09204.

*N-(4-((4-(3,4-Dichlorophenyl)thiazol-2-yl)amino)phenyl)isobutyramide* (**A**_30_): yellow solid; yield 71%; mp: 251.4–252.7 °C; ^1^H-NMR δ 10.21 (s, 1H, NH), 9.72 (s, 1H, NH), 8.10 (d, *J* = 1.9 Hz, 1H, Ar-H), 7.88 (m,1H, Ar-H), 7.65 (m, 1H, Ar-H), 7.55 (d, *J* = 9.7 Hz, Ar-H), 7.55–7.51 (m, 2H, Ar-H), 7.49 (s, 1H, thiazole-H), 2.54 (m, 1H, CH), 1.07 (d, *J* = 6.8 Hz, 6H). ^13^C-NMR δ 175.25, 164.11, 147.93, 136.84, 135.57, 134.09, 131.89, 131.39, 130.13, 127.63, 126.30, 120.49, 117.96, 105.31, 35.29, 20.05. HRMS (ESI): *m*/*z* [M + H]^+^ calculated for C_19_H_18_Cl_2_N_3_OS: 406.05421; found: 406.05307.

*N-(4-(4-(4-Bromophenyl)thiazol-2-ylamino)phenyl)isobutyramide* (**A**_31_): white solid; yield 87%; mp: 190.6–191.6 °C; ^1^H-NMR δ 10.19 (s, 1H, NH), 9.73 (s, 1H, NH), 7.92–7.82 (m, 2H, Ar-H), 7.66–7.60 (m, 4H, Ar-H), 7.57 (d, *J* = 9.2 Hz, 2H, Ar-H), 7.37 (s, 1H, thiazole-H), 2.65–2.53 (m, 1H, CH), 1.11 (d, *J* = 6.8 Hz, 6H, CH_3_). ^13^C-NMR δ 175.24, 163.98, 149.34, 137.00, 134.23, 133.97, 132.03, 128.17, 120.97, 120.50, 117.86, 103.91, 35.30, 20.05. HRMS (ESI): *m*/*z* [M + H]^+^ calculated for C_19_H_19_BrN_3_OS: 416.04267; found: 416.04153.

*N-(4-(4-(3-Bromophenyl)thiazol-2-ylamino)phenyl)isobutyramide* (**A**_32_)**:** white solid; yield 55%; mp: 238.2–239.0 °C; ^1^H-NMR δ 10.17 (s, 1H, NH), 9.70 (s, 1H, NH), 8.05 (d, *J* = 1.8 Hz, 1H, Ar-H), 7.90 (dd, *J* = 6.6, 1.2 Hz, 1H, Ar-H), 7.56 (d, *J* = 6.4 Hz, 2H, Ar-H), 7.54 (d, *J* = 6.4 Hz, 2H, Ar-H), 7.53–7.44 (m, 1H, Ar-H), 7.42 (s, 1H, thiazole-H), 7.36 (t, *J* = 7.9 Hz, 1H, Ar-H), 2.60–2.50 (m, 1H,CH), 1.07 (d, *J* = 6.8 Hz, 6H, CH_3_). ^13^C-NMR δ 175.25, 164.03, 148.81, 137.30, 136.94, 134.05, 131.35, 130.62, 128.58, 125.13, 122.56, 120.52, 117.94, 104.63, 35.30, 20.04. HRMS (ESI): *m*/*z* [M + H]^+^ calculated for C_19_H_19_BrN_3_OS: 416.04267; found: 416.04153.

*N-(4-((4-(4-(Trifluoromethyl)phenyl)thiazol-2-yl)amino)phenyl)isobutyramide* (**A**_33_): white solid; yield 71%; mp: 198.8–199.5 °C; ^1^H-NMR δ 10.23 (s, 1H, NH), 9.73 (s, 1H, NH), 8.24–8.13 (m, 2H, Ar-H), 7.67–7.61 (m, 2H, Ar-H), 7.60–7.56 (m, 2H, Ar-H), 7.55 (d, *J* = 6.4 Hz, 2H, Ar-H), 7.54 (s, 1H, thiazole-H), 2.58–2.50 (m, 1H, CH), 1.07 (d, *J* = 6.8 Hz, 6H, CH_3_). ^13^C-NMR δ 175.32, 164.27, 148.86, 136.96, 135.97, 134.18, 130.40, 130.03, 125.89, 124.45, 122.46, 120.55, 118.04, 105.04, 35.37, 20.11. HRMS (ESI): *m*/*z* [M + H]^+^ calculated for C_20_H_19_F_3_N_3_OS: 406.11954; found: 406.11868.

*N-(4-(4-(3-(Trifluoromethyl)phenyl)thiazol-2-ylamino)phenyl)isobutyramide* (**A**_34_): yellow solid; yield 81%; mp: 198.8–199.5 °C; ^1^H-NMR δ 10.21 (s, 1H), 9.71 (s, 1H), 8.19 (s, 2H, Ar-H), 7.64 (d, *J* = 6.7 Hz, 2H, Ar-H), 7.58 (m, 2H, Ar-H), 7.53 (s, 1H, thiazole-H), 2.59–2.50 (m, 1H, CH), 1.07 (d, *J* = 6.8 Hz, 6H, CH_3_). ^13^C-NMR δ 175.32, 164.30, 148.88, 136.98, 135.99, 134.20, 130.38, 130.19, 129.98, 125.90, 124.45, 123.73, 122.45, 120.58, 118.07, 105.02, 35.37, 20.10. HRMS (ESI): *m*/*z* [M + H]^+^ calculated for C_20_H_19_F_3_N_3_OS: 406.11954; found: 406.11826.

*N-(4-((4-(p-Tolyl)thiazol-2-yl)amino)phenyl)isobutyramide* (**A**_35_): yellow solid; yield 73%; mp: 245.2–246.4 °C; ^1^H-NMR δ 10.12 (s, 1H, NH), 9.71 (s, 1H, NH), 7.77 (d, *J* = 8.1 Hz, 2H, Ar-H), 7.58 (d, *J* = 9.1 Hz, 2H, Ar-H), 7.53 (d, *J* = 9.1 Hz, 2H, Ar-H), 7.20 (d, *J* = 8.0 Hz, 2H, Ar-H), 7.18 (s, 1H, thiazole-H), 2.62–2.50 (m, 1H, CH), 2.29 (s, 3H, CH_3_), 1.07 (d, *J* = 6.8 Hz, 6H, CH_3_). ^13^C-NMR δ 175.20, 163.70, 150.60, 137.20, 133.82, 132.41, 129.67, 126.09, 120.49, 117.72, 102.06, 35.28, 21.32, 20.07. HRMS (ESI): *m*/*z* [M + H]^+^ calculated for C_20_H_22_N_3_OS: 352.14781; found: 352.14685.

*N-(4-((4-Phenylthiazol-2-yl)amino)phenyl)isobutyramide* (**A**_36_): white solid; yield 84%; mp: 239.6–241.0 °C; ^1^H-NMR δ 10.15 (s, 1H, NH), 9.72 (s, 1H, NH), 7.88 (d, *J* = 7.2 Hz, 2H, Ar-H), 7.60 (d, *J* = 9.0 Hz, 2H, Ar-H), 7.55 (d, *J* = 9.0 Hz, 2H, Ar-H), 7.39 (t, *J* = 7.7 Hz, 2H, Ar-H), 7.28 (d, *J* = 7.3 Hz, 1H, Ar-H), 7.26 (s, 1H, Thiazole-H), 2.59–2.50 (m, 1H), 1.07 (d, *J* = 6.8 Hz, 6H, CH_3_). ^13^C-NMR δ 175.23, 150.54, 137.13, 135.04, 133.87, 129.11, 128.00, 126.15, 120.51, 117.76, 102.98, 35.29, 20.06. HRMS (ESI): *m*/*z* [M + H]^+^ calculated for C_19_H_20_N_3_OS: 338.13216; found: 338.13101.

### 3.3. X-ray Diffraction Analysis

To further confirm the three-dimensional structure of the target compounds, crystals (1.9 mm × 0.5 mm × 1.6 mm) of **A**_4_ (Figure 2) were obtained by slow evaporation and analyzed by X-ray diffraction. Cell dimensions and intensities were measured at 298 K on a Bruker Smart Apex CCD diffractometer with MoKα radiation (*λ* = 0.71073 Å). The structure was solved by direct method with the SHELXS-97 program. The results show that crystals of **A**_4_ is monoclinic system, which is characterized by no higher-order symmetry axis. A total of 2709 reflections were measured, of which 1714 were unique in the range of 2.19 < θ < 25.02° (h, −14 to 19; k, −6 to 5; l, −22 to 21). The two benzene rings in the molecule are almost in the same plane. All of the non-H atoms were refined anisotropically by full-matrix least-squares to give the final R = 0.0732 and WR = 0.1820. The completeness of the crystal data is 99.9%. The crystal data of the compound **A**_4_ have been deposited at the Cambridge Crystallographic Data Center under code CCDC 1,975,217 that contains all the above data.

### 3.4. In Vitro Antibacterial Activity Bioassays 

The in vitro antibacterial activity of target compounds against *Xoo*, *Xac* and *Xoc* was evaluated using a turbidimeter test [26,27]. For comparison, the commercial antibacterial agents bismerthiazol and thiodiazole copper was used as positive controls. DMSO in sterile water served as negative control. Nutrient broth medium (NB, 1 g of yeast powder, 3 g of beef extract, 10 g of glucose, 5 g of peptone and 1 L of distilled water, pH 7.0 to 7.2) in tubes was sterilized under high temperature and pressure. The tested compounds were dissolved in 120 μL DMSO then diluted with 0.1% Tween-20 solution, and finally working solution concentrations of 200 and 100 μg/mL were obtained. 1 mL of the solution containing compounds, bismerthiazol and thiadiazole copper was transferred into tubes (15 × 150 mm) containing 4 mL nutrient broth (NB) medium. Approximately 40 μL of activated bacteria was introduced into each tube. Finally, the test tubes were incubated at 28 ± 1 °C with continuous shaking at 180 rpm for 24–48h. The density value (OD_595_) of the solution in the tube was monitored on the microplates when the OD_595_ of the negative control group was 0.6 to 0.8. The inhibition rate was calculated using the following equation: Inhibitory Rate (%) = (CK − T)/CK × 100%.(1)
where “CK” represents the density value (OD_595_) of negative control group, and “T” implies the density value (OD_595_) of the treated NB medium. The EC_50_ values against *Xoo*, *Xac* and *Xoc* of the target compounds were tested at five gradient concentrations and computed from analysis using the SPSS 17.0 software (IBM, New York, NY, USA). The experiments were repeated three times for each compound.

### 3.5. Scanning Electron Microscopy

The cell surface was observed as previously described [28]. *Xoo* culture (OD_595_ = 1.0) was centrifuged and washed with PBS (pH = 7.2), then resuspended in 1.0 mL of PBS buffer. Then, the cells were treated with compound **A**_1_ for 10 h at 28 °C at concentrations of 50 μg/mL and 100 μg/mL. The untreated sample served as a negative control. Next, the compound solution was removed by washing three times with PBS buffer. The *Xoo* cells were fixed with 2.5% glutaraldehyde at 4 °C overnight and then dehydrated with 70%, and 90% ethanol, respectively.

### 3.6. Nematicidal Biological Activity In Vitro

The nematicidal activities of the compounds **A**_1_–**A**_36_ against *M. incognita* were tested by a typical assay [29,30]. The tested compounds were dissolved with DMSO and then diluted with 1% Tween-80 solution to prepare 500 and 100 μg/mL solutions. Approximately 200 μL of test solution was added into a 48 well plate. Then a suspension that included approximately 200 living nematodes was added into the above solution. Avermectin was used as positive control. The solution without compound was used as a negative control. All experiments were repeated three times. The mortality of the nematodes was seen under a stereoscopic binocular microscope after 24 and 72 h. The corrected mortality of nematicide was calculated using the following equation:Corrected Mortality (%) = (Mortality of Treatment − Mortality of Control)/(1 − Mortality of Control) × 100%

## 4. Conclusions

In summary, a series of new *N*-phenylacetamide derivatives containing 4-arylthiazole moieties was designed and synthesized by introducing the thiazole moiety into the scaffolds of *N*-phenylacetamides. The target compounds were evaluated for their in vitro antibacterial activities against *Xoo*, *Xac* and *Xoc*. The bactericidal activity data show that most of the target compounds have moderate inhibitory activities against the above pathogens, among which compounds **A**_1_ and **A**_3_ have good inhibitory effects on *Xoo* and *Xoc*. The cell-rupturing effect of compound **A**_1_ on *Xoo* was studied through SEM analysis. In addition their nematocidal activity was also determined and the compound **A**_23_ exhibited strong toxicity against *M. incognita*, comparable to that of avermectin.

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
