# Peer review of "Synthesis and Antibacterial Evaluation of N-phenylacetamide Derivatives Containing 4-Arylthiazole Moieties"

_molecules, 2020, doi:10.3390/molecules25081772_

Round 1
Reviewer 1 Report
The manuscript is accepted as it is
Reviewer 2 Report
The AA have now described in a more detailed and chemically clear manner the chemical procedure for the synthesis of their compounds.
Two observations:
-- the sentence “Among them, it was unexpectedly found that compounds A4, A7, A11, A12 have been reported” il rather surprising! Was does “unexpectedly” means? Maybe a more careful literature search should have been done from the very beginning! Anyway, the AA should report, in the experimental section, at least the physico-chemical data (melting point…) already reported for the known compounds, so as to confirm their own experimental data.
-- the English language is still very poor and I herein restate that an extensive editing (or a complete check by an English speaking person) should be done before publication.